

# Variation in the diversity and richness of parasitoid wasps based on sampling effort

Thomas E. Saunders[1] and Darren F. Ward[1,2]

[1] School of Biological Sciences, University of Auckland, Auckland, New Zealand
[2] Landcare Research, Auckland, New Zealand

Corresponding author
Darren F. Ward,
wardda@landcareresearch.co.nz

## ABSTRACT

Parasitoid wasps are a mega-diverse, ecologically dominant, but poorly studied component of global biodiversity. In order to maximise the efficiency and reduce the cost of their collection, the application of optimal sampling techniques is necessary. Two sites in Auckland, New Zealand were sampled intensively to determine the relationship between sampling effort and observed species richness of parasitoid wasps from the family Ichneumonidae. Twenty traps were deployed at each site at three different times over the austral summer period, resulting in a total sampling effort of 840 Malaise-trap-days. Rarefaction techniques and non-parametric estimators were used to predict species richness and to evaluate the variation and completeness of sampling. Despite an intensive Malaise-trapping regime over the summer period, no asymptote of species richness was reached. At best, sampling captured two-thirds of parasitoid wasp species present. The estimated total number of species present depended on the month of sampling and the statistical estimator used. Consequently, the use of fewer traps would have caught only a small proportion of all species (one trap 7–21%; two traps 13–32%), and many traps contributed little to the overall number of individuals caught. However, variation in the catch of individual Malaise traps was not explained by seasonal turnover of species, vegetation or environmental conditions surrounding the trap, or distance of traps to one another. Overall the results demonstrate that even with an intense sampling effort the community is incompletely sampled. The use of only a few traps and/or for very short periods severely limits the estimates of richness because (i) fewer individuals are caught leading to a greater number of singletons; and (ii) the considerable variation of individual traps means some traps will contribute few or no individuals. Understanding how sampling effort affects the richness and diversity of parasitoid wasps is a useful foundation for future studies.

## INTRODUCTION

Invertebrates comprise the vast majority of global species richness, drive wide-ranging ecological processes, and provide ecosystem services essential for human prosperity (*Mora et al., 2011*; *Chapin et al., 2000*; *Potts et al., 2010*). Despite their obvious importance, there are numerous impediments to a wider appreciation of their diversity, ecology,

and taxonomy (*Sluys, 2013*; *Wheeler et al., 2012*). A key barrier is the lack of accurate, standardised, and cost-effective sampling regimes to optimise the collection of species (*New, 1998*; *Keating et al., 1998*). Consequently, insects are commonly overlooked in conservation planning and assessment chiefly because obtaining even a basic understanding of their taxonomic and functional diversity is often difficult (*New, 2009*; *Samways, 2005*, *Holwell & Andrew, 2014*).

Parasitoid wasps are solitary insects whose larvae eventually kill the arthropod hosts they feed on during development (*Eggleton & Gaston, 1990*). Their total global diversity is estimated at over 350,000 species (*Gaston, 1991*), making them 7× more diverse than vertebrates. Parasitoid wasps are known to exert considerable influence over ecological processes in both natural and human-modified environments by regulating their host populations (*LaSalle & Gauld, 1993*). For example, parasitoids have repeatedly demonstrated their economic value to humans as biological control agents that significantly reduce the damage and economic losses caused by insect pests in agroecosystems (*Van Driesche et al., 2010*; *LaSalle & Gauld, 1993*). However, parasitoids can also be highly prone to extinction because of their reliance on hosts in lower trophic levels (*Shaw & Hochberg, 2001*).

Despite their ecological and economic significance, parasitoid wasps (and Hymenoptera in general) are disproportionately under-represented in insect conservation (*Shaw & Hochberg, 2001*; *Ward et al., 2012*). One of the best ways to include invertebrates in conservation assessment is to gain a better understanding of their distributions and diversity by comparing and evaluating a range of sampling regimes (*New, 2012*; *Southwood & Henderson, 2000*). Ultimately a cost-effective sampling program that returns an accurate inventory of species richness needs to: (i) capture the largest number of individuals in the least amount of time, (ii) predict how the number, spacing, and timing of trap deployments (sampling effort) affects observed species richness, and (iii) measure the completeness of samples collected to decide whether further sampling is worth the effort (*New, 1998*). This approach to surveying is termed 'optimal sampling', and despite being well developed for marine and freshwater invertebrates, it remains lacking for most groups of terrestrial invertebrates (*Basualdo, 2011*).

Malaise traps are widely considered to be the best method for capturing flying insects, particularly parasitoid wasps (*Van Achterberg, 2009*; *Noyes, 1989*). As passive flight interception traps, they can be left in the field for long periods and are simple to operate (*New, 1998*). However, despite their popularity, very little is known about how their number, placement, and sampling duration influence the species diversity of the resulting catch (*Fraser, Dytham & Mayhew, 2008*). Malaise traps are typically deployed in very low numbers which increases the risk of significantly underestimating total species richness, missing many rare taxa, or biasing the catch due to their placement (*Fraser, Dytham & Mayhew, 2007*). This can be problematic when few or low numbers of individuals are caught, because the number of individuals captured affects the accuracy of species diversity estimates (*Chao & Chiu, 2016*).

In addition, little is known about how numbers and placement of Malaise traps, and the duration of trap effort, influences the species richness and diversity of observed parasitoid
wasps. In one of the few studies to examine sampling effort, *Fraser, Dytham & Mayhew (2008)* intensively studied ichneumonid wasps from two sites in the UK. Even when employing 16 Malaise traps per site, and catching large proportions of all UK species (28%) from a single site, overall the parasitoid community was incompletely sampled. Consequently, whilst using a small number of traps can contribute useful information about the parasitoid community, their use is likely to greatly underestimate total species richness.

The aim of this paper was to investigate how variation in sampling effort (number of traps, duration) within a site affects the observed and estimated species richness of parasitoid wasps (Hymenoptera: Ichneumonidae). Such information will inform sampling strategies to facilitate cost-effective evaluations of parasitoid wasp diversity.

## MATERIALS AND METHODS

### Study sites

This study was undertaken within the Waitakere Ranges, located in northern New Zealand. The Waitakere Ranges encompasses 17,000 ha, ranging from sea level to 474 m, with mean annual temperature from 12.5 to 14.5 °C and mean annual precipitation between 1,400 and 2,000 mm (*Jongkind & Buurman, 2006*). The vegetation is comprised of a podocarp-broadleaf mixture, with aggregated stands of the endemic conifer, Kauri (*Agathis australis*).

Two sites within the greater Waitakere Ranges were selected for sampling, Huapai (36°47′43.8″S, 174°29′25.5″E) and Oratia (36°55′01.3″S, 174°36′12.0″E). Permission to conduct the study was obtained through the Auckland Council (#CS57). Both sites are characterised as Kauri forest in various stages of regeneration (*Thomas & Ogden, 1983*). The mean temperature over the study period was 17.7 °C (lowest 11.8 °C, highest 25.1 °C). Total rainfall was 71 mm.

### Sampling

Twenty Malaise traps were used at each site. Traps were variants of the *Townes (1972)* style, predominantly the 'ez-Malaise trap' (http://bugdorm.megaview.com). Traps were spaced 20–30 m along a central transect that ran approximately through the middle of each site, and some traps were haphazardly placed 20–30 m either side of the central transect. Where possible, traps were positioned across likely flight paths of insects with the head facing the sun's zenith (*Noyes, 1989*; *Van Achterberg, 2009*).

Sampling took place during the austral summer in 2014–2015, and three sampling periods were conducted at each site, each of one week duration. Huapai was sampled between 24 November–1 December 2014, 7–14 January 2015, and 2–9 February 2015. Oratia was sampled between 2–9 December 2014, 17–24 January 2015, and 10–17 February 2015. This gave a total of 420 Malaise-trap-days for each site (20 Malaise traps, three sampling periods, and seven days per trap). Collection bottles were filled with a 50/50 mix of ethanol and glycol.

All Hymenoptera were removed from Malaise trap samples, and stored in 95% ethanol. Ichneumonidae were sorted to subfamily and genus level. Specimens were identified primarily through an online key to genera (*Schnitzler & Ward, 2013*). The New Zealand ichneumonid fauna is relatively poorly known, with most groups requiring extensive
species-level revision. For taxa lacking resolution to species level, a morphospecies approach was used. To improve the accuracy of delimitation, morphospecies were compared with specimens within the New Zealand Arthropod Collection. DNA barcoding was utilised to ensure females and males were correctly associated (Supplementary Information 1). All sequences are publicly available on BOLD (http://www.boldsystems.org) under the project 'IchneumonidaeDiversityMSc (LKMSC)' with Process Id numbers from LKMSC191-16 to LKMSC285-16. All specimens are held at the New Zealand Arthropod Collection, Auckland.

## Environmental factors

In order to determine the influence of vegetation and environmental variables on Hymenoptera catches, at each trap a circular quadrat of 5 m diameter was used to measure: (i) the total number of pieces of coarse woody debris (CWD) (with a diameter of at least 7 cm); (ii) the number of all plant species present; and (iii) proportional measures of five ground cover types: Kauri debris (*A. australis*); Nikau Palm (*Rhopalostylis sapida*) debris; CWD; other leaf litter; and bare ground.

## Data analysis

**Species estimates.** A range of species richness indices and estimators were generated from sample-based datasets using EstimateS v9.1.0 (*Colwell & Elsensohn, 2014*). Three richness estimators were used based on their accuracy under conditions relevant to the present study (for review see *Hortal, Borges & Gaspar, 2006*): two incidence-based estimators (ICE and Chao2) and the second-order 'jackknife' estimator (Jack2). The 'classic formula' was used for calculating Chao2 values as recommended in-software (*Colwell & Elsensohn, 2014*). A total of 1,000 randomizations were used for each analysis, and the upper abundance limit for rare or infrequent taxa was left at the default of 10. This parameter allows users to specify an upper limit for samples or individuals, below which the species would be classified as 'rare'. This information is used by EstimateS for calculating ACE and ICE (see *Chazdon et al., 1997*). Sampling efficiency was calculated from Observed species ($S_{obs}$)/Estimated species ($S_{est}$).

Sample-based taxon resampling (rarefaction) curves were generated from EstimateS output for each period at each site to show the relationship between sampling effort and observed species richness. Individual-based rarefaction curves were also plotted to provide a comparison of species richness that takes into account the different number of individuals collected from each site (*Gotelli & Colwell, 2001*).

**Species composition.** Similarity-based, multivariate analyses were conducted in PRIMER-E v.6 (*Clarke & Gorley, 2006*). Data did not require standardising as identical sampling procedures and effort were maintained over the course of the study. Parasitoid abundance data were square root transformed. Environmental data were normalised by subtracting the mean and dividing by the standard deviation within each variable. Factors were site (Oratia or Huapai); trap type (one of three Townes-style designs used); and sample month (December, January, February).

Resemblance matrices were constructed by applying a zero-adjusted Bray–Curtis coefficient to parasitoid abundance, and Euclidean distance to environmental data

**Table 1 Summary of Malaise trap catches of Ichneumonidae from Huapai and Oratia sites across the sampling periods (December, January, February, and combined).**

| Site/period (trap days) | Individuals | Observed species ($S_{obs}$) | Singletons | Doubletons | Singletons/$S_{obs}$ % |
|---|---|---|---|---|---|
| **Huapai** | | | | | |
| December (140) | 60 | 23 | 17 | 1 | 73.9 |
| January (140) | 113 | 29 | 10 | 5 | 34.5 |
| February (140) | 210 | 35 | 18 | 1 | 51.4 |
| Combined (**420**) | 383 | 49 | 18 | 7 | 36.7 |
| **Oratia** | | | | | |
| December (140) | 33 | 13 | 8 | 2 | 61.5 |
| January (140) | 88 | 18 | 12 | 0 | 66.7 |
| February (140) | 64 | 20 | 12 | 3 | 60.0 |
| Combined (**420**) | 185 | 33 | 19 | 1 | 57.6 |
| **Total (840)** | 568 | 61 | 24 | 10 | 39.3 |

Notes:
Number of individuals; number of species caught ($S_{obs}$); number of singletons; number of doubletons; and the ratio of singletons/$S_{obs}$.

(*Clarke & Gorley, 2006*). A zero-adjusted Bray–Curtis coefficient addresses the 'double zero' problem and helps to correct the erratic behaviour of the coefficient when samples become sparse (*Clarke, Somerfield & Chapman, 2006*). In PRIMER-E, zero-adjusting the coefficient is accomplished by adding a 'dummy variable' with values of 1 for each sample (*Clarke & Gorley, 2006*). This method was justified because sample sizes were sufficiently large and were taken close enough together in space to assume that samples would have low species richness and abundance for the same reason. For non-metric Multi-Dimensional Scaling (nMDS) ordination, 50 restarts with a minimum stress value of 0.01 were used for each analysis.

**Environmental factors.** Physical distances between traps were measured (range 20–180 m apart) and the similarity of the catch (using Bray–Curtis similarity coefficient) was determined for each pair of traps (20 traps, 190 pairwise combinations) to examine spatial autocorrelation. A BEST test was used to compare environmental variables surrounding a trap with the parasitoid composition and abundance (*Clarke & Gorley, 2006*).

## RESULTS

### Parasitoid diversity

A total sampling effort of 840 Malaise-trap-days at both sites over a period of three months resulted in the capture of 61 morphospecies, from 568 individuals. Only 10% of species were identified to the species level. There was a very strong correlation ($R = 0.934$) between frequency (number of times sampled) and abundance (number of individuals) of species.

Despite the intensive sampling effort, only 13 species comprised more than 10 individuals, while 40% of species were singletons ($n = 24$) and 16% were doubletons ($n = 10$). Each month there was a high ratio of singletons (known from only one sample) to observed species richness (range 34.5–73.9%; Table 1). The number of

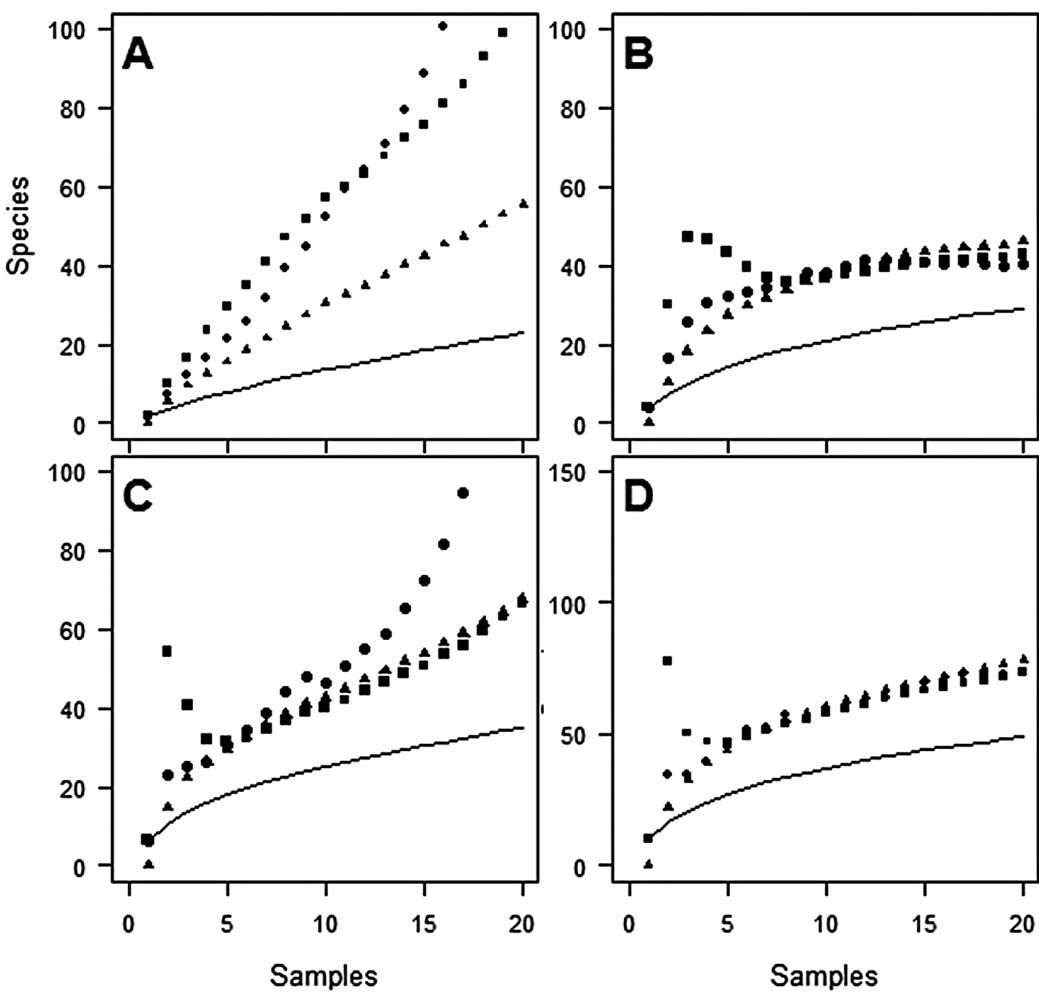

**Figure 1 Species accumulation curves at Huapai for each sampling period.** (A) December; (B) January; (C) February; (D) Combined periods, comparing observed species richness (solid line) with three non-parametric estimators of total species richness (ICE—squares; Jack2—triangles; Chao2—circles).

ichneumonids caught was <1 per trap per day (all periods; Haupai = 0.91 per day; Oraita = 0.44 per day), except Haupai in February (Table 1).

Observed species richness ($S_{obs}$) peaked in February at both sites (Table 1). A greater number of species and individuals were sampled at the Haupai site, and this was consistent each month. Twenty-eight species were found only at the Huapai site (46%), 14 were exclusive to Oratia (23%), and 19 were found at both localities (31%).

## Species accumulation

Despite the use of 20 traps, observed species richness failed to reach an asymptote at either site (Figs. 1 and 2). Sampling efficiency ($S_{obs}/S_{est}$) was only 62.7–66.9% at Huapai and 36.6–50.3% at Oratia, depending on the species richness estimator used.

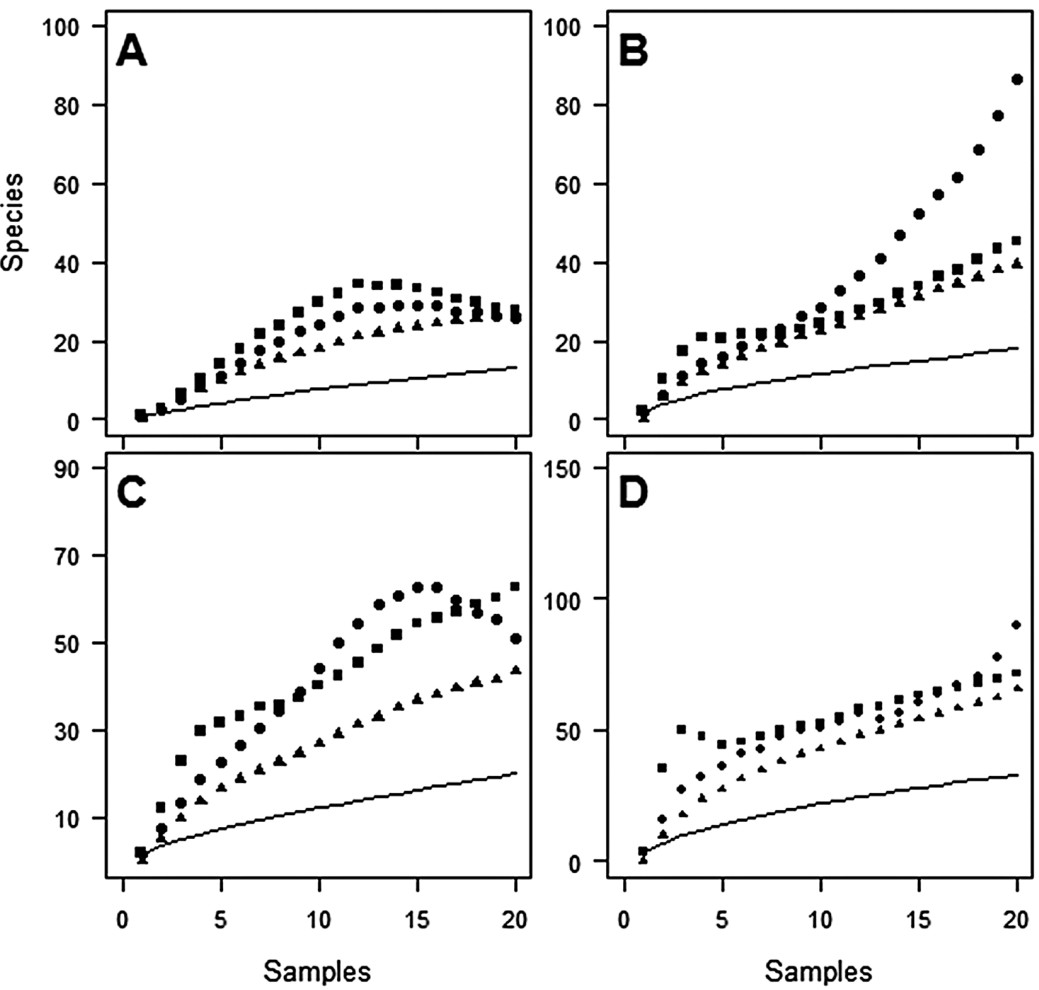

**Figure 2 Species accumulation curves at Oratia for each sampling period.** (A) December; (B) January; (C) February; (D) Combined periods, comparing observed species richness (solid line) with three non-parametric estimators of total species richness (ICE—squares; Jack2—triangles; Chao2—circles).

The average proportion of species caught from one trap varied between 7.2% and 21.2%, and from two traps between 13.2% and 32.4%, of the total catch from 20 traps (depending on site/period combinations, see Table 2).

Estimates of species richness were often very different depending on the month of sampling (Table 2). For example, using the ICE estimator, estimates of species richness at Huapai varied between 105 species (December), 42 species (January), and 66 species (February).

Not surprisingly, more individuals were captured with greater sampling effort, that is, in combinations of two or three periods. However, examining combinations of different time periods could be useful for planning about specifically when to undertake sampling. For both sites, a combination of January and February resulted in the capture of the highest number of individuals (Fig. 3). However, the combination of December and

**Table 2 Estimated species richness of Ichneumonidae from Malaise trap catches at Huapai and Oratia sites across the sampling periods (December, January, February, and combined) using three estimators (ICE, Jack2, and Chao2).**

| Site/Period | ICE | Jack2 | Chao2 | 1 Trap% | 2 Traps% |
|---|---|---|---|---|---|
| **Huapai** | | | | | |
| December | 105.8 | 55.4 | 176.9 | 8.6 | 16.3 |
| January | 42.8 | 46.0 | 40.4 | 14.0 | 24.6 |
| February | 66.7 | 67.4 | 188.9 | 18.3 | 30.0 |
| Combined | 73.2 | 78.1 | 73.5 | 21.2 | 32.4 |
| **Oratia** | | | | | |
| December | 27.8 | 27.0 | 25.8 | 7.2 | 13.2 |
| January | 45.2 | 39.3 | 86.4 | 11.4 | 20.6 |
| February | 62.5 | 43.3 | 51.0 | 9.4 | 17.7 |
| Combined | 71.4 | 65.5 | 90.1 | 11.9 | 21.0 |
| **Total** | 93.9 | 98.7 | 90.6 | 19.7 | 31.0 |

**Note:**
Proportions of species caught from 1 trap to 2 traps (as a % from a total of 20 traps).

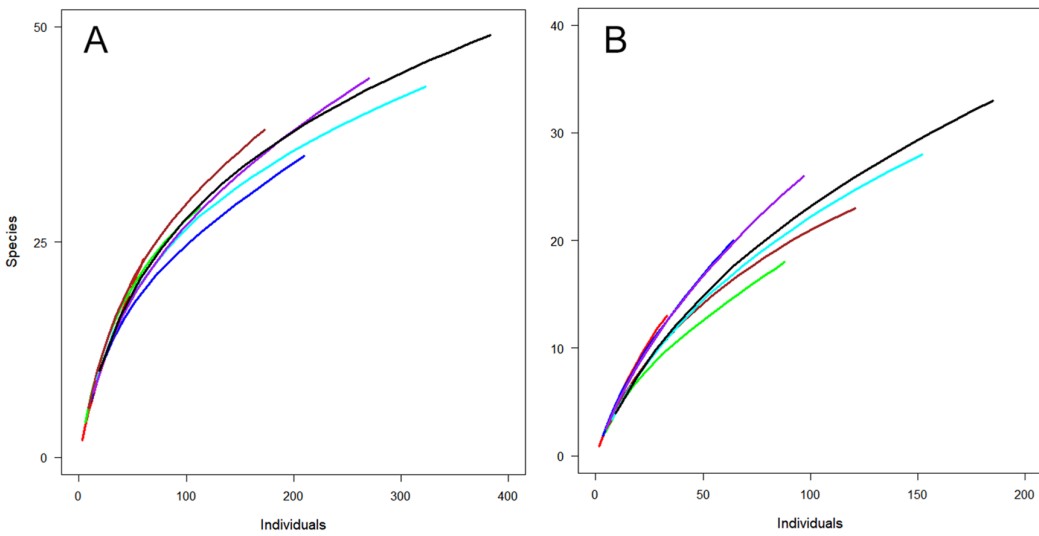

**Figure 3 Observed species richness for individual-based rarefaction curves for Huapai (A) and Oratia (B) for different sampling periods.** December (red); January (green); February (dark blue); December + January (dark red); December + February (light blue); January + February (purple); All periods (black).

February captured a similar number of species from fewer individuals (Fig. 3), indicating the importance of sampling in December (early summer).

## Variation of individual traps

Some traps contributed very little to the numbers of overall individuals caught (Table 3). The top 10 traps (i.e. 50% of traps) caught 81% of individuals at Huapai and 93% of individuals at Oratia (Table 3). However, at Oratia 20% of traps ($n = 4$) failed to catch any

**Table 3 Variation in the catches of Ichneumonidae from all periods of summer using Malaise traps at Huapai and Oratia.**

|  | Number traps with no catch | Number traps with 10 or less individuals | Number of individuals (%) of top five traps | Number of individuals (%) of top 10 traps |
|---|---|---|---|---|
| Huapai | 0 | 7 | 191 (50%) | 312 (81%) |
| Oratia | 4 | 13 | 127 (69%) | 172 (93%) |

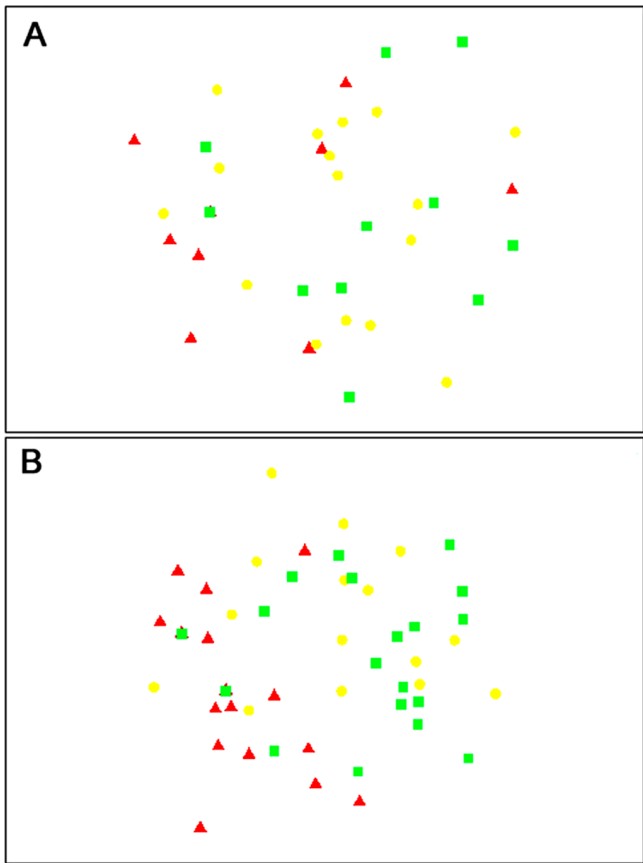

**Figure 4 nMDS plots showing similarity of parasitoid assemblages.** Huapai (A) and Oratia (B) at different sampling periods: December (red triangle), January (yellow circle), and February (green square).                   

ichneumonids over the entire sampling period. Furthermore, one-third of traps at Huapai ($n = 7/20$), and two-thirds at Oratia ($n = 13/20$) caught less than 10 individuals.

The composition of parasitoids was not affected by site (ANOSIM $R = 0.161$, Fig. 4A) or month (ANOSIM $R = 0.109$, Fig. 4B). The distances between sets of traps (pairwise distances) did not affect the similarity of the trap catches (using Bray–Curtis similarity coefficient) for either Huapai (Rho = $-0.366$, Fig. 5A), or Oratia (Rho = $-0.338$, Fig. 5B). Furthermore, there was no relationship between environmental factors (vegetation structure, plant diversity, ground cover) immediately surrounding a trap and the composition of parasitoids in the trap (BEST analysis, Rho = 0.18).

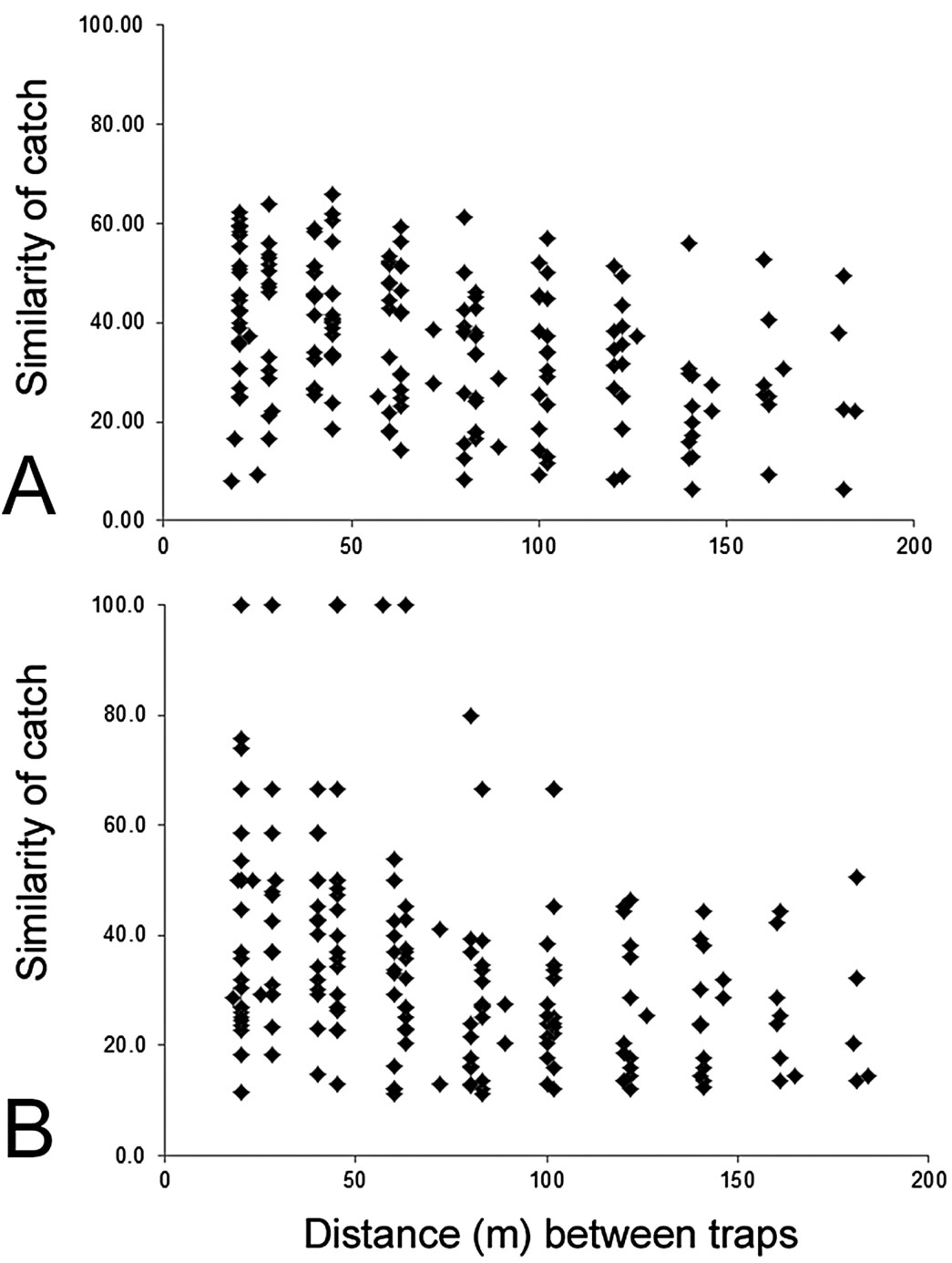

**Figure 5 Similarity of trap catches (Bray–Curtis similarity coefficient) against the distance between each trap combinations.** Huapai (A) and Oratia (B).

## DISCUSSION

Our aim of this paper was to investigate how variation in sampling effort affects the observed and estimated species richness of Ichneumonid parasitoid wasps.

We found that even with an intensive Malaise-trapping regime covering a three-month summer period, at best only two-thirds of parasitoid wasp species were captured

(approximately 62–67% of species were captured at Haupai, but less, 36–50%, at Oratia). There was also a high degree of variation in estimated number of species depending on the month of sampling and the statistical estimator used. Consequently, the use of only one or two traps would have caught only a small proportion of all species. Furthermore, many traps contributed little to the overall number of individuals caught. Variation in the catch of individual Malaise traps was not explained by seasonal turnover of species, vegetation or environmental conditions surrounding the trap, or the distance of traps to one another.

A number of recent studies have examined the biology of ichneumonid parasitoid wasps and how their diversity relates to such factors as the influence of vegetation structure on life history traits (*Saaksjarvi et al., 2006*); differences between habitat types (*Mazon & Bordera, 2008*; *Kendall & Ward, 2016*); altitudinal gradients (*Hall et al., 2015*); diversity in the tropics and along latitudinal gradients (*Eagalle & Smith, 2017*); and spillover of species between natural and managed forests (*Frost et al., 2015*). However, very few studies have examined how numbers and placement of Malaise traps and the duration of trap effort influences the species richness and diversity of parasitoid wasps (*Darling & Packer, 1988*; *Fraser, Dytham & Mayhew, 2008*).

*Fraser, Dytham & Mayhew (2008)* found that although local parasitoid diversity can be very high from a single site, for example, capturing 28% of all UK species, even with high sampling effort the overall parasitoid community was still incompletely sampled. They showed how a small number of traps underestimated total species richness, for example, two traps would have caught 32–47% of the total species (*Fraser, Dytham & Mayhew, 2008*). Our study confirmed this general finding, but found that two traps would have captured an even lower proportion of species (13–32%). Furthermore, we showed that individual trap catch was highly variable. Remarkably, 20% of traps ($n = 4$) at Oratia failed to catch any ichneumonids over the entire sampling period. Additionally, one-third of traps at Huapai, and two-thirds at Oratia caught less than 10 individuals in total.

The failure to reach an asymptote in the accumulation of species is typical of most invertebrate surveys, even those employing intensive sampling effort sustained over long periods of time (*Coddington et al., 2009*; *Longino, Coddington & Colwell, 2002*). The explanation of 'rarity' (e.g. singletons) is one key to understanding how differences in sampling methods and effort affects estimates of species richness (*Novotný & Basset, 2000*; *Longino, Coddington & Colwell, 2002*). Excessive numbers of singletons present statistical problems, as non-parametric estimators use the prevalence of rare species in a set of samples to calculate the true number of species in the habitat: as the ratio of singletons to doubletons increases, so do the estimates of species richness (*Gotelli & Colwell, 2001*). There was a high degree of variation in estimated number of species depending on the month of sampling and the statistical estimator used. Part of this variation relates to the statistical uncertainty about the range of likely species contained within the samples. When 'fewer' samples are present, estimates of richness are often larger, but once there are 'sufficient' samples, more accurate estimates of richness are generated, and these are not always the highest estimate. However, all estimators depend on the variability

of the samples, and our data contains many samples with zero individuals and few samples with many individuals. The use of statistical estimators has had considerable review (*Hortal, Borges & Gaspar, 2006*; *Colwell & Elsensohn, 2014*), and the three estimators we choose (ICE, Chao2, Jack2) were based on such reviews, and all are commonly employed in ecology.

High numbers of singleton species are especially common in arthropod surveys, and they may result from errors or biases of sampling methodology (*Coddington et al., 2009*; *Novotný & Basset, 2000*). While Malaise traps are extensively used to catch flying insects, and particularly Ichneumonidae (*Van Achterberg, 2009*; *Noyes, 1989*), they are only one of several collecting techniques that could be used (*New, 1998*). Understanding whether a species is caught (or not caught) by a specific sampling technique is important to understand their occurrence and abundance in an area. If a species is present but the sampling method and does not catch or 'detect' a species, then this will bias estimates of species richness and rarity. The combined use of different sampling techniques is vital to fully understand species richness and diversity; however, this comes this an added cost of time to collect and process larger numbers of samples.

*Longino, Coddington & Colwell (2002)* completed an inventory of ant diversity over 14 years in Costa Rica. Despite their enormous sampling effort, 51 species (12% of the total) were still singletons (known from only one sample) at the end of the inventory. They were able to show why many species remained singletons because their long-term study provided excellent knowledge of the rarity of the fauna based on geography, methodology, and local and global distributions. In our study the parasitoid community was characterised by a high proportion of singletons (40%), however, we only utilised one sampling method. More complete inventories of communities are possible if multiple sampling methods and extensive effort are applied (*Longino, Coddington & Colwell, 2002*).

An alternative explanation is that some of these singletons are 'tourist' species (*Novotný & Basset, 2000*), passing through the study site between patches of resources (*Frost et al., 2015*). However, the landscapes surrounding the study sites are a mixture of heavily managed agricultural and urban land-use types, and all of the Ichneumonid species caught in this study are endemic (restricted to New Zealand) or native species, and so are likely to be resident within the native study sites, and not passing through from the surrounding modified landscapes. However, a fundamental limitation of our interpretation of overall diversity and rarity is the very poor knowledge of the ichneumonid fauna in New Zealand. There is a basic lack of knowledge regarding host association, habitat preferences, and geographical distribution for the majority of species (*Ward, 2012*; *Kendall & Ward, 2016*).

## CONCLUSION AND RECOMMENDATIONS

Biodiversity surveys should include some measure of their completeness to contextualise findings and facilitate comparison with other surveys (*New, 1998*). Quantification of sampling effort in relation to observed species richness is often the first step in developing standardised, cost-effective, optimal sampling regimes (*New, 1998*).

This study quantified how sampling effort affects the observed and estimated species richness of parasitoid wasps (Hymenoptera: Ichneumonidae) in New Zealand for the first time. Understanding how sampling effort affects the richness and diversity of parasitoid wasps is a useful first step for future studies. Overall, the results demonstrate that sampling effort (number of trap used, duration) strongly influences the observed and estimated species richness of parasitoid wasps. Malaise traps work well for the collection of Ichneumonidae, but far more trapping effort is needed; the use of only a few traps and/or for very short periods limits the estimates of richness because (i) fewer individuals are caught with a greater number of singletons; and (ii) the considerable variation of individual traps means some traps will contribute few or no individuals.

## ACKNOWLEDGEMENTS

We thank those who helped in the field, and staff from Landcare Research and the New Zealand Arthropod Collection for their support and expertise.

### Funding

The Centre for Biodiversity & Biosecurity provided funding to conduct molecular sequencing. This project was supported through Ministry of Business, Innovation & Employment funding to Landcare Research within the Characterising New Zealand's Land Biota Portfolio. The funders had no role in study design, data collection and analysis, decision to publish, or preparation of the manuscript.

### Grant Disclosures

The following grant information was disclosed by the authors:
The Centre for Biodiversity & Biosecurity.
Ministry of Business, Innovation & Employment.

### Competing Interests

Darren Ward is an Academic Editor for PeerJ and an employee of Landcare Research.

### Author Contributions

- Thomas E. Saunders conceived and designed the experiments, performed the experiments, analysed the data, contributed reagents/materials/analysis tools, prepared figures and/or tables, authored or reviewed drafts of the paper, approved the final draft.
- Darren F. Ward conceived and designed the experiments, performed the experiments, analysed the data, contributed reagents/materials/analysis tools, prepared figures and/or tables, authored or reviewed drafts of the paper, approved the final draft.

### Field Study Permissions

The following information was supplied relating to field study approvals (i.e. approving body and any reference numbers):
  Field experiments were approved by the Auckland Council Permit #CS57.

## Data Availability

The raw data are provided in Supplemental Dataset Files.

## Supplemental Information

Supplemental information for this article can be found online at http://dx.doi.org/10.7717/peerj.4642#supplemental-information.

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
