# Peer review of "Variation in the diversity and richness of parasitoid wasps based on sampling effort"

_PeerJ, doi:10.7717/peerj.4642_

## Round 0.1 · original submission · Minor Revisions

The reviewers all concur that this is a well-written paper with sound design and analysis. However, reviewers also concur that more effort is required in the discussion/conclusions section, where the paper falls short in explaining why these results are important in a wider context. I will be looking for evidence that this has been done seriously in the revised manuscript.

·

Basic reporting

The manuscript is clear on the main questions that are being answered by this study, and although the writing is a little clunky in places, the paper is quite concise overall.
One section that I would like to have more detail on is where the barcoded material was submitted, as these should be linked to this publication, and direct other users to identify the morpho-species in the same manner.

Experimental design

The study largely describes the outcomes of an exploratory pilot study, in a very poorly studied system so I believe the sampling design was suitable.

Validity of the findings

The value here is clarity on the significant effort that would be required to characterize biodiversity of this group. Given that despite substantial effort, sampling was still inadequate to compare sites or seasons there is little more that can be said, and indeed the authors do not overly speculate.

Additional comments

I think this kind of information is valuable to help any future studies calibrate their expectations when designing studies, especially if they wish to target specialist groups like this study did. On the other hand, of course, given how incomplete the records appeared to be, its difficult to guess how much more time sampling is needed.
The only main flaw I felt the paper has is that it focuses on species rarity as the reason for species richness accumulating so slowly. However, the total number of individuals caught was very low as acknowledged, and therefore this is actually an issue of "detectability" (see reviews like Guillera-Arroita, 2016). Whilst Malaise traps may have been more successful elsewhere, they may not in fact be well-suited to capturing these species, and you acknowledge that other kinds of traps may have increased the rate of species' recovery. The proportion of species that are then hard to observe because they are rare can only really be judged in light of that. This distinction doesn't really alter the study's finding or conclusions, but I would still recommend the authors consider updating a few sections of the discussion to reflect these differing perspectives.

·

Basic reporting

The manuscript is well written and clear (or rather, what is presented is clear; there are significant gaps (see below). Figures are well designed, and data were easy to access.

Experimental design

The research question is well thought out and presented. The work appears to have been well executed. As to methods: PeerJ wants “Methods described with sufficient detail to replicate”. That is not done here; but that doesn’t matter, because that is not actually how scientists write, or should write. (Replication detail belongs in online supplements, or at protocols.io, etc; it doesn’t belong in papers where it makes readers suffer. Of course, I realize that not all will agree with me on this.) What *does* matter is enough detail for readers to understand and interpret the results; and here, this manuscript falls short. See under “General comments” for specific examples of where we need to know more.

Validity of the findings

I believe the results are robust and well analyzed, although given shortcomings in the Methods I’m unable to be completely sure. The Discussion fell flat, to me; there was absolutely nothing said about why the diversity estimators behaved so differently, or about whether power constraints account for the lack of effects of environment. The paper would be much more useful with that; the current version essentially ends with “we didn’t find many of the species and aren’t sure by what fraction we underestimate diversity”. It’s possible to do more.

Additional comments

By line number:

57 “numerous impediments block” – a bit awkward. What about “there are numerous impediments to”?

67 Is this known global diversity or likely global diversity? Seems very low for the latter; make clear. And I’d be tempted to say “much more diverse” than vertebrates, or maybe give a vertebrate diversity number, to make a stronger point here (vertebrates ~50,000 max, I think?)

117ff I would encourage the authors to rewrite the Methods (at least) in the active voice. It’s shorter, more engaging, clearer; we are moving past our passive-voice convention.

125 I’d like to see coordinates for the two study sites, not just for the broader study area.

133 What were the other traps, what differences were there between trap types, and did you take this into account in the analysis? Were trap types assigned randomly to sites/locations?

135 I doubt that trap placement was “random”; I suspect it was actually “haphazard”. This is fine; just describe it as such! If I’m wrong and it actually was random – how was randomization accomplished? Randomly generated coordinates?

136 What is a “flight path” for a parasitoid wasp, and how were such identified?

137 I find this a bit unsatisfying without knowing anything about the microhabitats in question. What were they, how were they identified, how many traps in each, etc?

152 No methods for DNA barcoding? How was it done, was it done for every specimen or just for ones where they was doubt, did you always find a database match, what did you do with the sequences, etc etc?

158-160 This is a little superficial. What is Nikau? Why is number of CWD the right metric, not estimated total volume (two short pieces aren’t twice as much CWD as one long piece)? If none of the cover types involve living plants, then why did you also need to identify the plant species present? Did you identify all plants present or just dominants?

171 I think it would be worth specifying what the software does with this parameter – does it mean that anything with fewer than 10 observations is ignored? That would have a huge effect. Does it mean something else?

182 I don’t understand this – the environmental data weren’t factors, so how were they involved? I assume they were, since you standardized them. Ah, I see in the next paragraph, some kind of “Euclidean distance to environmental data” – but I don’t understand that. Also note, you can’t standardize environmental variable (iii) because “the plant species present” doesn’t have a mean or standard deviation – unless you mean the *number* of plant species present?

193 I’m not familiar with this technical sense of “denuded” and I wonder if other readers might be like me.

199 I’m not sure what relationship this has to 187. Are they redundant?

208 I assume you mean morphospecies? Otherwise, “only 10% of the species were identified to the species level” doesn’t make much sense. In the Discussion, you should probably comment on the likelihood that the true diversity is higher than the number of morphospecies, since there are cryptic species in parasitoids.

209 If you are reporting a correlation, you want the correlation coefficient (r), not the coefficient of determination (R2).

220-221 How many of these were singletons and doubletons? I don’t think it means much if some species were caught at only one site, if there were only a few captures of them – in other words, a site X captures G-test would not be significant.

225 (Figs 1-2). I don’t know enough about the estimators, I guess, because I don’t understand how they can be other than strictly increasing as you add more samples. ICE seems especially problematic in this sense. I hope in the Discussion this is something that will be addressed.

229 I don’t understand this. From which two traps? Random pairs of traps? For all pairs of traps, as at 198?

236 If this means what it seems to, isn’t this result completely trivial? Of course individuals for Jan + individuals for Feb > individuals for Jan; mathematically it can only be equal or greater. I would simply delete this, unless I’m missing its meaning (in which case, rephrase).

266 The differences among estimators need much more discussion. For the same data, there was a two-fold difference among estimators. Not only that: at least one (ICE) gave counterintuitive results in which, for some sampling intensities, adding more samples led to *lower* diversity estimates. So: what is known about the different performance of these estimators? Is one known to be most conservative? How do their assumptions differ, and does the assumption set of one estimator fit your situation better? How might one proceed to gather further, or different, data to resolve these differences? I would see this one phrase, “and the statistical estimator used”, expanded to at least one long-ish paragraph, if not more.

269 It would be interesting to know a bit more about the power you had to detect such effects if they were present. This is phrased appropriately – you can’t reject the null – but it would be useful to know whether you can place any bounds on the effect sizes. You may not be able to do a lot here, but I suspect given the very high inter-trap variability, power is low. Which is part of your point! Can you, for example, estimate variability for the traps that are most similar in terms of environment, and use that to suggest how strong an environmental effect would have to be to stand out against that background?

302 Since there were still many singletons, and thus many unsampled species, the inventory was not “exhaustive” (although it was probably exhausting!).

314 I don’t think this follows at all. Just because a species is native to NZ does not mean that it will be resident in every habitat type. In fact, if you thought this was true, you would not have attempted to estimate effects of environment on diversity or species representation in your trapping data! I would simply delete this sentence. Your singletons could very well be “tourists”.

Reviewer 3 ·

Basic reporting

All metrics of "basic reporting" meet high quality standards.

Experimental design

The research reported answers a well defined research question that has relevant and meaningful impacts on our understanding of (what is likely to be) perhaps one of the most diverse families of Metazoans in the world.

I found that the experimental design, statistical analyses and overall questions were all performed very capably.

Validity of the findings

I found that the experimental design, statistical analyses and overall questions were all performed very capably.

One suggestion I would make is that the authors elaborate briefly in the Conclusions and Recommendations on the importance that the continued deployment of Malaise traps has for the communities understanding of ichneumonid diversity (if, they in fact feel this way). To me, I worry that a cursory reading or reporting of this study once published might be that, “Malaise traps don’t work” when in fact, MORE Malaise trapping is needed (and people to sort their contents, if we are ever to more fully understand this diverse family!

Additional comments

The authors investigated how variation in sampling effort affected the observed and estimated species richness of Ichneumonid parasitoid wasps. While it is “well known” that ichneumonids are both diverse and often captured in Malaise sampling, there is a lack of a synthetic analysis of how well the common use of this common trap type actually captures the diversity of organisms present. Turns out, not well.

I have a couple of minor suggestions that I would suggest they deal with prior top publication.

SPECIFIC SUGGESTIONS

Line 249 – “3-month summer period”. To someone unfamiliar with a New Zealand summer – what proportion of the “flight period” might this be?

Line 260 – “Our aim….” Rather than “the aim of this paper. “

Line 270 (ish) – perhaps useful to insert some comment here regarding the likely even greater under-representation of total diversity in the tropics.

Line 270 ish – We recently (https://doi.org/10.1139/facets-2016-0061) used Malaise trapped parasitoids to test for anomalous latitudinal patterns of diversity in parasitic Hymenoptera. We noted that in no single site, did any of the diversity and sampling relationships reach an asymptote.

Line 302 – “14 years” rather than “many”.

Line 152: “DNA barcoding was utilised to ensure females and males were correctly associated.” –I have two further suggestions regarding this sentence. The first is that you report the number of times that this strategy was completed/necessary. Readers outside of the immediate taxonomic community would likely be unaware of how frequently such information is unknown and that such a strategy would be required. The second recommendation I have is to report the DNA barcode BIN (Barcode Index Numbers) associated with the species you including your matrix. These BINS serve as interim species name proxies until (and if) the morphospecies you delineate here are formally described. The frequency of interim species epithets is unavoidable in many arthropod groups (and in parasitoids specifically). However, I would strongly urge you to avoid the use of “sp.1” as it conveys no information outside of the current study. The unique BIN names can serve this purpose. If not all “species” have been barcoded and have BIN details, then I would suggest using the surname of the taxonomist to create an alpha numeric (NAME001). This will at least permit future readers and users of your data to consider and contextualise your findings. Otherwise, they join the long queue of locally specific and incomparable “sp.1” ‘s in the world!

---

## Round 0.2 · accepted · Accept

You have provided clear and comprehensive responses to the reviewers' questions, and I am happy to recommend your manuscript for publication - well done!

#